# Behaviour Suite for Reinforcement Learning

Ian Osband,* Yotam Doron, Matteo Hessel, John Aslanides
Eren Sezener, Andre Saraiva, Katrina McKinney, Tor Lattimore, Csaba Szepesvari
Satinder Singh, Benjamin Van Roy, Richard Sutton, David Silver, Hado Van Hasselt

**DeepMind**

## Abstract

This paper introduces the *Behaviour Suite for Reinforcement Learning*, or `bsuite` for short. `bsuite` is a collection of carefully-designed experiments that investigate core capabilities of reinforcement learning (RL) agents with two objectives. First, to collect clear, informative and scalable problems that capture key issues in the design of general and efficient learning algorithms. Second, to study agent behaviour through their performance on these shared benchmarks. To complement this effort, we open source `github.com/deepmind/bsuite`, which automates evaluation and analysis of any agent on `bsuite`. This library facilitates reproducible and accessible research on the core issues in RL, and ultimately the design of superior learning algorithms. Our code is Python, and easy to use within existing projects. We include examples with OpenAI Baselines, Dopamine as well as new reference implementations. Going forward, we hope to incorporate more excellent experiments from the research community, and commit to a periodic review of `bsuite` from a committee of prominent researchers.

## 1 Introduction

The reinforcement learning (RL) problem describes an agent interacting with an environment with the goal of maximizing cumulative reward through time (Sutton & Barto, 2017). Unlike other branches of control, the dynamics of the environment are not fully known to the agent, but can be learned through experience. Unlike other branches of statistics and machine learning, an RL agent must consider the effects of its actions upon future experience. An efficient RL agent must address three challenges simultaneously:

1. **Generalization:** be able to learn efficiently from data it collects.
2. **Exploration**: prioritize the right experience to learn from.
3. **Long-term consequences**: consider effects beyond a single timestep.

The great promise of reinforcement learning are agents that can learn to solve a wide range of important problems. According to some definitions, an agent that can learn to perform at or above human level across a wide variety of tasks is an artificial general intelligence (AGI) (Minsky, 1961; Legg et al., 2007).

Interest in artificial intelligence has undergone a resurgence in recent years. Part of this interest is driven by the constant stream of innovation and success on high profile challenges previously deemed impossible for computer systems. Improvements in image recognition are a clear example of these accomplishments, progressing from individual digit recognition (LeCun et al., 1998), to mastering ImageNet in only a few years (Deng et al., 2009; Krizhevsky et al., 2012). The advances in RL systems have been similarly impressive: from checkers (Samuel, 1959), to Backgammon (Tesauro, 1995), to Atari games (Mnih et al., 2015a), to competing with professional players at DOTA (Pachocki et al., 2019) or StarCraft (Vinyals et al., 2019) and beating world champions at Go (Silver et al., 2016). Outside of playing games, decision systems are increasingly guided by AI systems (Evans & Gao, 2016).

---

*Corresponding author iosband@google.com.

As we look towards the next great challenges for RL and AI, we need to understand our systems better (Henderson et al., 2017). This includes the scalability of our RL algorithms, the environments where we expect them to perform well, and the key issues outstanding in the design of a general intelligence system. We have the existence proof that a single self-learning RL agent can master the game of Go purely from self-play (Silver et al., 2018). We do not have a clear picture of whether such a learning algorithm will perform well at driving a car, or managing a power plant. If we want to take the next leaps forward, we need to continue to enhance our understanding.

## 1.1 PRACTICAL THEORY OFTEN LAGS PRACTICAL ALGORITHMS

The practical success of RL algorithms has built upon a base of theory including gradient descent (Bottou, 2010), temporal difference learning (Sutton, 1988) and other foundational algorithms. Good theory provides insight into our algorithms beyond the particular, and a route towards general improvements beyond ad-hoc tinkering. As the psychologist Kurt Lewin said, 'there is nothing as practical as good theory' (Lewin, 1943). If we hope to use RL to tackle important problems, then we must continue to solidify these foundations. This need is particularly clear for RL with nonlinear function approximation, or 'deep RL'. At the same time, theory often lags practice, particularly in difficult problems. We should not avoid practical progress that can be made before we reach a full theoretical understanding. The successful development of algorithms and theory typically moves in tandem, with each side enriched by the insights of the other.

The evolution of neural network research, or *deep learning*, provides a poignant illustration of how theory and practice can develop together (LeCun et al., 2015). Many of the key ideas for deep learning have been around, and with successful demonstrations, for many years before the modern deep learning explosion (Rosenblatt, 1958; Ivakhnenko, 1968; Fukushima, 1979). However, most of these techniques remained outside the scope of developed learning theory, partly due to their complex and non-convex loss functions. Much of the field turned away from these techniques in a 'neural network winter', focusing instead of function approximation under convex loss (Cortes & Vapnik, 1995). These convex methods were almost completely dominant until the emergence of benchmark problems, mostly for image recognition, where deep learning methods were able to clearly and objectively demonstrate their superiority (LeCun et al., 1998; Krizhevsky et al., 2012). It is only now, several years after these high profile successes, that learning theory has begun to turn its attention back to deep learning (Kawaguchi, 2016; Bartlett et al., 2017; Belkin et al., 2018). The current theory of deep RL is still in its infancy. In the absence of a comprehensive theory, the community needs principled benchmarks that help to develop an understanding of the strengths and weaknesses of our algorithms.

## 1.2 AN 'MNIST' FOR REINFORCEMENT LEARNING

In this paper we introduce the *Behaviour Suite for Reinforcement Learning* (or `bsuite` for short): a collection of experiments designed to highlight key aspects of agent scalability. Our aim is that these experiments can help provide a bridge between theory and practice, with benefits to both sides. These experiments embody fundamental issues, such as 'exploration' or 'memory' in a way that can be easily tested and iterated. For the development of theory, they force us to instantiate measurable and falsifiable hypotheses that we might later formalize into provable guarantees. While a full theory of RL may remain out of reach, the development of clear experiments that instantiate outstanding challenges for the field is a powerful driver for progress. We provide a description of the current suite of experiments and the key issues they identify in Section 2.

Our work on `bsuite` is part of a research process, rather than a final offering. We do not claim to capture all, or even most, of the important issues in RL. Instead, we hope to provide a simple library that collects the best available experiments, and makes them easily accessible to the community. As part of an ongoing commitment, we are forming a `bsuite` committee that will periodically review the experiments included in the official `bsuite` release. We provide more details on what makes an 'excellent' experiment in Section 2, and on how to engage in their construction for future iterations in Section 5.

The Behaviour Suite for Reinforcement Learning is a not a replacement for 'grand challenge' undertakings in artificial intelligence, or a leaderboard to climb. Instead it is a collection of diagnostic experiments designed to provide *insight* into key aspects of agent behaviour. Just as the MNIST dataset offers a clean, sanitised, test of image recognition as a stepping stone to advanced computer vision; so too `bsuite` aims to instantiate targeted experiments for the development of key RL capabilities.

The successful use of illustrative benchmark problems is not unique to machine learning, and our work is similar in spirit to the *Mixed Integer Programming Library* (MIPLIB) (miplib2017). In mixed integer programming, and unlike linear programming, the majority of algorithmic advances have (so far) eluded theoretical analysis. In this field, MIPLIB serves to instantiate key properties of problems (or types of problems), and evaluation on MIPLIB is a typical component of any new algorithm. We hope that `bsuite` can grow to perform a similar role in RL research, at least for those parts that continue to elude a unified theory of artificial intelligence. We provide guidelines for how researchers can use `bsuite` effectively in Section 3.

### 1.3 Open source code, reproducible research

As part of this project we open source `github.com/deepmind/bsuite`, which instantiates all experiments in code and automates the evaluation and analysis of any RL agent on `bsuite`. This library serves to facilitate reproducible and accessible research on the core issues in reinforcement learning. It includes:

- Canonical implementations of all experiments, as described in Section 2.
- Reference implementations of several reinforcement learning algorithms.
- Example usage of `bsuite` with alternative codebases, including 'OpenAI Gym'.
- Launch scripts for Google cloud that automate large scale compute at low cost.[1]
- A ready-made `bsuite` Jupyter notebook with analyses for all experiments.
- Automated LaTeX appendix, suitable for inclusion in conference submission.

We provide more details on code and usage in Section 4.

We hope the Behaviour Suite for Reinforcement Learning, and its open source code, will provide significant value to the RL research community, and help to make key conceptual issues concrete and precise. `bsuite` can highlight bottlenecks in *general* algorithms that are not amenable to hacks, and reveal properties and scalings of algorithms outside the scope of current analytical techniques. We believe this offers an avenue towards great leaps on key issues, separate to the challenges of large-scale engineering (Nair et al., 2015). Further, `bsuite` facilitates clear, targeted and unified experiments across different code frameworks, something that can help to remedy issues of reproducibility in RL research (Tanner & White, 2009; Henderson et al., 2017).

### 1.4 Related work

The Behaviour Suite for Reinforcement Learning fits into a long history of RL benchmarks. From the beginning, research into general learning algorithms has been grounded by the performance on specific environments (Sutton & Barto, 2017). At first, these environments were typically motivated by small MDPs that instantiate the general learning problem. 'CartPole' (Barto et al., 1983) and 'MountainCar' (Moore, 1990) are examples of classic benchmarks that has provided a testing ground for RL algorithm development. Similarly, when studying specific capabilities of learning algorithms, it has often been helpful to design diagnostic environments with that capability in mind. Examples of this include 'RiverSwim' for exploration (Strehl & Littman, 2008) or 'Taxi' for temporal abstraction (Dietterich, 2000). Performance in these environments provide a targeted signal for particular aspects of algorithm development.

As the capabilities or RL algorithms have advanced, so has the complexity of the benchmark problems. The Arcade Learning Environment (ALE) has been instrumental in driving

---

[1]At August 2019 pricing, a full `bsuite` evaluation for our DQN implementation cost under $6.

progress in deep RL through surfacing dozens of Atari 2600 games as learning environments (Bellemare et al., 2013). Similar projects have been crucial to progress in continuous control (Duan et al., 2016; Tassa et al., 2018), model-based RL (Wang et al., 2019) and even rich 3D games (Beattie et al., 2016). Performing well in these complex environments requires the integration of many core agent capabilities. We might think of these benchmarks as natural successors to 'CartPole' or 'MountainCar'.

The Behaviour Suite for Reinforcement Learning offers a complementary approach to existing benchmarks in RL, with several novel components:

1. `bsuite` experiments enforce a specific *methodology* for agent evaluation beyond just the environment definition. This is crucial for scientific comparisons and something that has become a major problem for many benchmark suites (Machado et al., 2017) (Section 2).
2. `bsuite` aims to isolate core capabilities with targeted 'unit tests', rather than integrate general learning ability. Other benchmarks evolve by increasing complexity, `bsuite` aims to remove all confounds from the core agent capabilities of interest (Section 3).
3. `bsuite` experiments are designed with an emphasis on *scalability* rather than final performance. Previous 'unit tests' (such as 'Taxi' or 'RiverSwim') are of fixed size, `bsuite` experiments are specifically designed to vary the complexity smoothly (Section 2).
4. `github.com/deepmind/bsuite` has an extraordinary emphasis on the ease of use, and compatibility with RL agents not specifically designed for `bsuite`. Evaluating an agent on `bsuite` is practical even for agents designed for a different benchmark (Section 4).

## 2 EXPERIMENTS

This section outlines the experiments included in the Behaviour Suite for Reinforcement Learning 2019 release. In the context of `bsuite`, an *experiment* consists of three parts:

1. **Environments**: a fixed set of environments determined by some parameters.
2. **Interaction**: a fixed regime of agent/environment interaction (e.g. 100 episodes).
3. **Analysis**: a fixed procedure that maps agent *behaviour* to results and plots.

One crucial part of each `bsuite` analysis defines a 'score' that maps agent performance on the task to $[0, 1]$. This score allows for agent comparison 'at a glance', the Jupyter notebook includes further detailed analysis for each experiment. All experiments in `bsuite` only measure *behavioural* aspects of RL agents. This means that they only measure properties that can be observed in the environment, and are not internal to the agent. It is this choice that allows `bsuite` to easily generate and compare results across different algorithms and codebases. Researchers may still find it useful to investigate internal aspects of their agents on `bsuite` environments, but it is not part of the standard analysis.

Every current and future `bsuite` experiment should target some key issue in RL. We aim for simple behavioural experiments, where agents that implement some concept well score better than those that don't. For an experiment to be included in `bsuite` it should embody five key qualities:

- **Targeted**: performance in this task corresponds to a key issue in RL.
- **Simple**: strips away confounding/confusing factors in research.
- **Challenging**: pushes agents beyond the normal range.
- **Scalable**: provides insight on scalability, not performance on one environment.
- **Fast**: iteration from launch to results in under 30min on standard CPU.

Where our current experiments fall short, we see this as an opportunity to improve the Behaviour Suite for Reinforcement Learning in future iterations. We can do this both through replacing experiments with improved variants, and through broadening the scope of issues that we consider.

We maintain the full description of each of our experiments through the code and accompanying documentation at `github.com/deepmind/bsuite`. In the following subsections, we pick two `bsuite` experiments to review in detail: 'memory length' and 'deep sea', and review these examples in detail. By presenting these experiments as examples, we can emphasize what we think makes `bsuite` a valuable tool for investigating core RL issues. We do provide a high level summary of all other current experiments in Appendix A.

To accompany our experiment descriptions, we present results and analysis comparing three baseline algorithms on `bsuite`: DQN (Mnih et al., 2015a), A2C (Mnih et al., 2016) and Bootstrapped DQN (Osband et al., 2016). As part of our open source effort, we include full code for these agents and more at `bsuite/baselines`. All plots and analysis are generated through the automated `bsuite` Jupyter notebook, and give a flavour for the sort of agent comparisons that are made easy by `bsuite`.

## 2.1 EXAMPLE EXPERIMENT: MEMORY LENGTH

Almost everyone agrees that a competent learning system requires *memory*, and almost everyone finds the concept of memory intuitive. Nevertheless, it can be difficult to provide a rigorous definition for memory. Even in human minds, there is evidence for distinct types of 'memory' handled by distinct regions of the brain (Milner et al., 1998). The assessment of memory only becomes more difficult to analyse in the context of general learning algorithms, which may differ greatly from human models of cognition. Which types of memory should we analyse? How can we inspect belief models for arbitrary learning systems? Our approach in `bsuite` is to sidestep these debates through simple behavioural experiments.

We refer to this experiment as *memory length*; it is designed to test the number of sequential steps an agent can remember a single bit. The underlying environment is based on a stylized T-maze (O'Keefe & Dostrovsky, 1971), parameterized by a length $N \in \mathbb{N}$. Each episode lasts $N$ steps with observation $o_t = (c_t, t/N)$ for $t = 1, .., N$ and action space $\mathcal{A} = \{-1, +1\}$. The context $c_1 \sim \text{Unif}(\mathcal{A})$ and $c_t = 0$ for all $t \geq 2$. The reward $r_t = 0$ for all $t < N$, but $r_N = \text{Sign}(a_N = c_1)$. For the `bsuite` experiment we run the agent on sizes $N = 1, .., 100$ exponentially spaced and look at the average regret compared to optimal after 10k episodes. The summary 'score' is the percentage of runs for which the average regret is less than 75% of that achieved by a uniformly random policy.

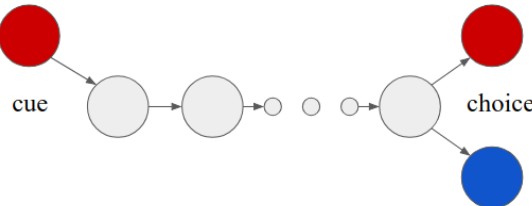

Figure 1: Illustration of the 'memory length' environment

*Memory length* is a good `bsuite` experiment because it is targeted, simple, challenging, scalable and fast. By construction, an agent that performs well on this task has mastered some use of memory over multiple timesteps. Our summary 'score' provides a quick and dirty way to compare agent performance at a high level. Our sweep over different lengths $N$ provides empirical evidence about the scaling properties of the algorithm beyond a simple pass/fail. Figure 2a gives a quick snapshot of the performance of baseline algorithms. Unsurprisingly, actor-critic with a recurrent neural network greatly outperforms the feedforward DQN and Bootstrapped DQN. Figure 2b gives us a more detailed analysis of the same underlying data. Both DQN and Bootstrapped DQN are unable to learn anything for length $> 1$, they lack functioning memory. A2C performs well for all $N \leq 30$ and essentially random for all $N > 30$, with quite a sharp cutoff. While it is not surprising that the recurrent agent outperforms feedforward architectures on a memory task, Figure 2b gives an excellent insight into the scaling properties of this architecture. In this case, we have a clear explanation for the observed performance: the RNN agent was trained via backprop-through-time with length 30. `bsuite` recovers an empirical evaluation of the scaling properties we would expect from theory.

## 2.2 EXAMPLE EXPERIMENT: DEEP SEA

Reinforcement learning calls for a sophisticated form of exploration called *deep exploration* (Osband et al., 2017). Just as an agent seeking to 'exploit' must consider the long term

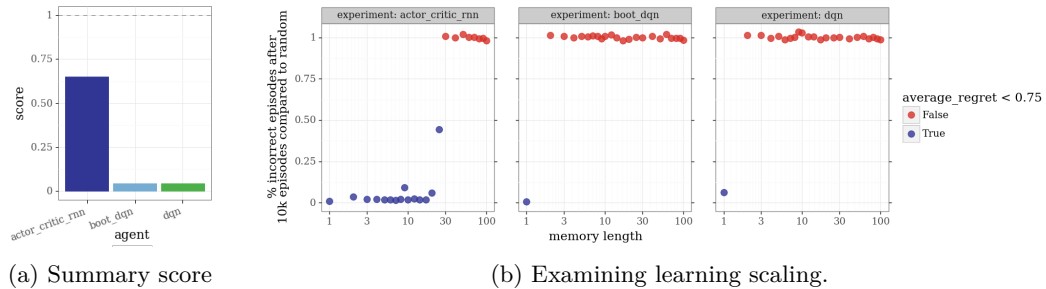

(a) Summary score

(b) Examining learning scaling.

Figure 2: Selected output from `bsuite` evaluation on 'memory length'.

consequences of its actions towards cumulative rewards, an agent seeking to 'explore' must consider how its actions can position it to learn more effectively in future timesteps. The literature on efficient exploration broadly states that only agents that perform deep explo- ration can expect polynomial sample complexity in learning (Kearns & Singh, 2002). This literature has focused, for the most part, on uncovering possible strategies for deep explo- ration through studying the tabular setting analytically (Jaksch et al., 2010; Azar et al., 2017). Our approach in `bsuite` is to complement this understanding through a series of behavioural experiments that highlight the need for efficient exploration.

The deep sea problem is implemented as an $N \times N$ grid with a one-hot encoding for state. The agent begins each episode in the top left corner of the grid and descends one row per timestep. Each episode terminates after $N$ steps, when the agent reaches the bottom row. In each state there is a random but fixed mapping between actions $\mathcal{A} = \{0, 1\}$ and the transitions 'left' and 'right'. At each timestep there is a small cost $r = -0.01/N$ of moving right, and $r = 0$ for moving left. However, should the agent transition right at every timestep of the episode it will be rewarded with an additional reward of $+1$. This presents a particularly challenging exploration problem for two reasons. First, following the 'gradient' of small intermediate rewards leads the agent *away* from the optimal policy. Second, a policy that explores with actions uniformly at random has probability $2^{-N}$ of reaching the rewarding state in any episode. For the `bsuite` experiment we run the agent on sizes $N = 10, 12, .., 50$ and look at the average regret compared to optimal after 10k episodes. The summary 'score' computes the percentage of runs for which the average regret drops below 0.9 faster than the $2^N$ episodes expected by dithering.

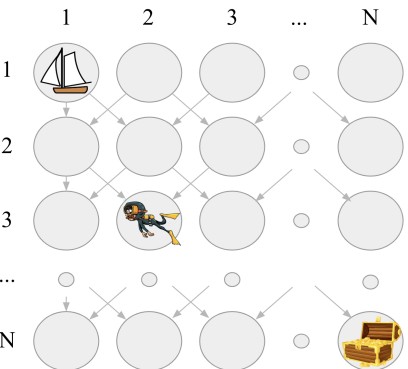

Figure 3: Deep-sea exploration: a simple example where deep exploration is critical.

*Deep Sea* is a good `bsuite` experiment because it is targeted, simple, challenging, scalable and fast. By construction, an agent that performs well on this task has mastered some key properties of deep exploration. Our summary score provides a 'quick and dirty' way to compare agent performance at a high level. Our sweep over different sizes $N$ can help to pro- vide empirical evidence of the scaling properties of an algorithm beyond a simple pass/fail. Figure 3 presents example output comparing A2C, DQN and Bootstrapped DQN on this

task. Figure 4a gives a quick snapshot of performance. As expected, only Bootstrapped DQN, which was developed for efficient exploration, scores well. Figure 4b gives a more detailed analysis of the same underlying data. When we compare the scaling of learning with problem size $N$ it is clear that only Bootstrapped DQN scales gracefully to large problem sizes. Although our experiment was only run to size 50, the regular progression of learning times suggest we might expect this algorithm to scale towards $N > 50$.

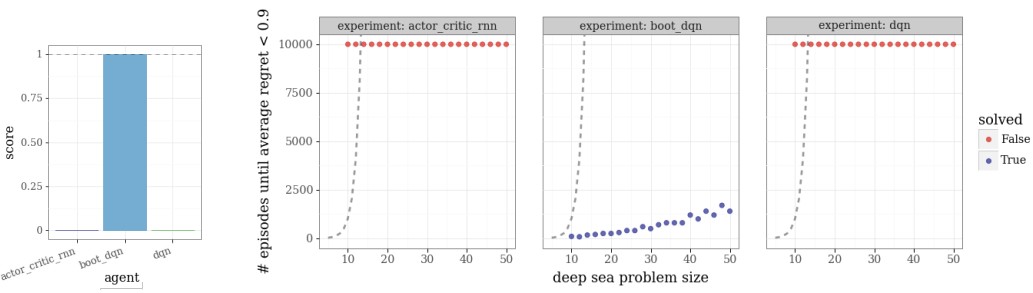

(a) Summary score     (b) Examining learning scaling. Dashed line at $2^N$ for reference.

Figure 4: Selected output from `bsuite` evaluation on 'deep sea'.

## 3  How to use `bsuite`

This section describes some of the ways you can use `bsuite` in your research and development of RL algorithms. Our aim is to present a high-level description of some research and engineering use cases, rather than a tutorial for the code installation and use. We provide examples of specific investigations using `bsuite` in Appendixes C, D and E. Section 4 provides an outline of our code and implementation. Full details and tutorials are available at `github.com/deepmind/bsuite`.

A `bsuite` experiment is defined by a set of environments and number of episodes of interaction. Since loading the environment via `bsuite` handles the logging automatically, any agent interacting with that environment will generate the data required for required for analysis through the Jupyter notebook we provide (Pérez & Granger, 2007). Generating plots and analysis via the notebook only requires users to provide the path to the logged data. The 'radar plot' (Figure 5) at the start of the notebook provides a snapshot of agent behaviour, based on summary scores. The notebook also contains a complete description of every experiment, summary scoring and in-depth analysis of each experiment. You can interact with the full report at `bit.ly/bsuite-agents`.

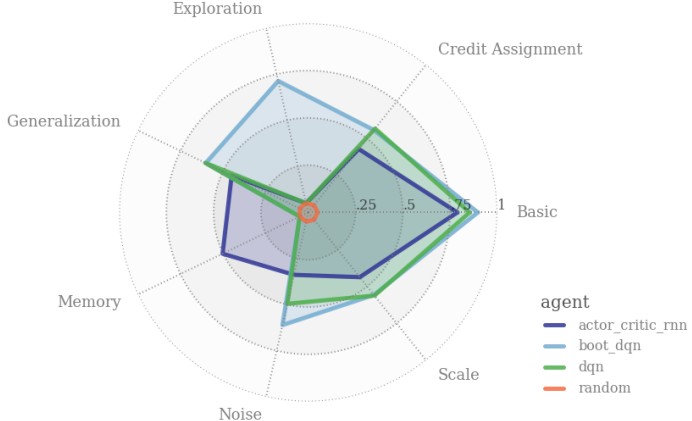

Figure 5: We aggregate experiment performance with a snapshot of 7 core capabilities.

If you are developing an algorithm to make progress on fundamental issues in RL, running on `bsuite` provides a simple way to replicate benchmark experiments in the field. Although

many of these problems are 'small', in the sense that their solution does not necessarily require large neural architecture, they are designed to highlight key challenges in RL. Further, although these experiments do offer a summary 'score', the plots and analysis are designed to provide much more information than just a leaderboard ranking. By using this common code and analysis, it is easy to benchmark your agents and provide reproducible and verifiable research.

If you are using RL as a tool to crack a 'grand challenge' in AI, such as beating a world champion at Go, then taking on `bsuite` gridworlds might seem like small fry. We argue that one of the most valuable uses of `bsuite` is as a diagnostic 'unit-test' for large-scale algorithm development. Imagine you believe that 'better exploration' is key to improving your performance on some challenge, but when you try your 'improved' agent, the performance does not improve. Does this mean your agent does not do good exploration? Or maybe that exploration is not the bottleneck in this problem? Worse still, these experiments might take days and thousands of dollars of compute to run, and even then the information you get might not be targeted to the key RL issues. Running on `bsuite`, you can test key capabilities of your agent and diagnose potential improvements much faster, and more cheaply. For example, you might see that your algorithm completely fails at credit assignment beyond $n = 20$ steps. If this is the case, maybe this lack of credit-assignment over long horizons is the bottleneck and not necessarily exploration. This can allow for much faster, and much better informed agent development - just like a good suite of tests for software development.

Another benefit of `bsuite` is to disseminate your results more easily and engage with the research community. For example, if you write a conference paper targeting some improvement to hierarchical reinforcement learning, you will likely provide some justification for your results in terms of theorems or experiments targeted to this setting.[2] However, it is typically a large amount of work to evaluate your algorithm according to alternative metrics, such as exploration. This means that some fields may evolve without realising the connections and distinctions between related concepts. If you run on `bsuite`, you can automatically generate a one-page Appendix, with a link to a notebook report hosted online. This can help provide a scientific evaluation of your algorithmic changes, and help to share your results in an easily-digestible format, compatible with ICML, ICLR and NeurIPS formatting. We provide examples of these experiment reports in Appendices B, C, D and E.

## 4   CODE STRUCTURE

To avoid discrepancies between this paper and the source code, we suggest that you take practical tutorials directly from `github.com/deepmind/bsuite`. A good starting point is `bit.ly/bsuite-tutorial`: a Jupyter notebook where you can play the code right from your browser, without installing anything. The purpose of this section is to provide a high-level overview of the code that we open source. In particular, we want to stress is that `bsuite` is designed to be a library for RL research, not a framework. We provide implementations for all the environments, analysis, run loop and even baseline agents. However, it is not necessary that you make use of them all in order to make use of `bsuite`.

The recommended method is to implement your RL agent as a class that implements a `policy` method for action selection, and an `update` method for learning from transitions and rewards. Then, simply pass your agent to our run loop, which enumerates all the necessary `bsuite` experiments and logs all the data automatically. If you do this, then all the experiments and analysis will be handled automatically and generate your results via the included Jupyter notebook. We provide examples of running these scripts locally, and via Google cloud through our tutorials.

If you have an existing codebase, you can still use `bsuite` without migrating to our run loop or agent structure. Simply replace your environment with `environment = bsuite.load_and_record(bsuite_id)` and add the flag `bsuite_id` to your code. You can then complete a full `bsuite` evaluation by iterating over the `bsuite_ids` defined in

---

[2]A notable omission from the `bsuite`2019 release is the lack of any targeted experiments for 'hierarchical reinforcement learning' (HRL). We invite the community to help us curate excellent experiments that can evaluate quality of HRL.

`sweep.SWEEP`. Since the environments handle the logging themselves, your don't need any additional logging for the standard analysis. Although full `bsuite` includes many separate evaluations, no single `bsuite` environment takes more than 30 minutes to run and the sweep is naturally parallel. As such, we recommend launching in parallel using multiple processes or multiple machines. Our examples include a simple approach using Python's `multiprocessing` module with Google cloud compute. We also provide examples of running bsuite from OpenAI baselines (Dhariwal et al., 2017) and Dopamine (Castro et al., 2018).

Designing a single RL agent compatible with diverse environments can cause problems, particularly for specialized neural networks. `bsuite` alleviates this problem by specifying an `observation_spec` that surfaces the necessary information for adaptive network creation. By default, `bsuite` environments implement the `dm_env` standards (Muldal et al., 2017), but we also include a wrapper for use through Openai `gym` (Brockman et al., 2016). However, if your agent is hardcoded for a format, `bsuite` offers the option to output each environment with the `observation_spec` of your choosing via linear interpolation. This means that, if you are developing a network suitable for Atari and particular `observation_spec`, you can choose to swap in `bsuite` without any changes to your agent.

## 5 Future iterations

This paper introduces the Behaviour Suite for Reinforcement Learning, and marks the start of its ongoing development. With our opensource effort, we chose a specific collection of experiments as the `bsuite2019` release, but expect this collection to evolve in future iterations. We are reaching out to researchers and practitioners to help collate the most informative, targeted, scalable and clear experiments possible for reinforcement learning. To do this, submissions should implement a `sweep` that determines the selection of environments to include and logs the necessary data, together with an `analysis` that parses this data.

In order to review and collate these submissions we will be forming a `bsuite` committee. The committee will meet annually during the NeurIPS conference to decide which experiments will be included in the `bsuite` release. We are reaching out to a select group of researchers, and hope to build a strong core formed across industry and academia. If you would like to submit an experiment to `bsuite` or propose a committee member, you can do this via github pull request, or via email to `bsuite.committee@gmail.com`.

We believe that `bsuite` can be a valuable tool for the RL community, and particularly for research in deep RL. So far, the great success of deep RL has been to leverage large amounts of computation to improve performance. With `bsuite`, we hope to leverage large-scale computation for improved *understanding*. By collecting clear, informative and scalable experiments; and providing accessible tools for reproducible evaluation we hope to facilitate progress in reinforcement learning research.

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

## A    Experiment summary

This appendix outlines the experiments that make up the `bsuite` 2019 release. In the interests of brevity, we provide only an outline of each experiment here. Full documentation for the environments, interaction and analysis are kept with code at `github.com/deepmind/bsuite`.

### A.1    Basic learning

We begin with a collection of very simple decision problems, and standard analysis that confirms an agent's competence at learning a rewarding policy within them. We call these experiments 'basic', since they are not particularly targeted at specific core issues in RL, but instead test a general base level of competence we expect all general agents to attain.

#### A.1.1    Simple bandit

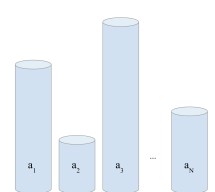

| component | description |
| --- | --- |
| **environments** | Finite-armed bandit with deterministic rewards $[0, 0.1, ..1]$ (Gittins, 1979). 20 seeds. |
| **interaction** | 10k episodes, record regret vs optimal. |
| **score** | regret normalized [random, optimal] $\rightarrow$ [0,1] |
| **issues** | basic |

#### A.1.2    MNIST

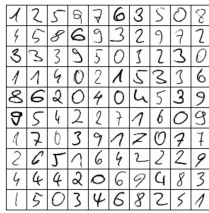

| component | description |
| --- | --- |
| **environments** | Contextual bandit classification of MNIST with $\pm 1$ rewards (LeCun et al., 1998). 20 seeds. |
| **interaction** | 10k episodes, record average regret. |
| **score** | regret normalized [random, optimal] $\rightarrow$ [0,1] |
| **issues** | basic, generalization |

#### A.1.3    Catch

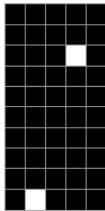

| component | description |
| --- | --- |
| **environments** | A 10x5 Tetris-grid with single block falling per column. The agent can move left/right in the bottom row to 'catch' the block. 20 seeds. |
| **interaction** | 10k episodes, record average regret. |
| **score** | regret normalized [random, optimal] $\rightarrow$ [0,1] |
| **issues** | basic, credit assignment |

#### A.1.4    Cartpole

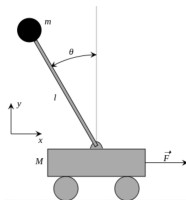

| component | description |
| --- | --- |
| **environments** | Agent can move a cart left/right on a plane to keep a balanced pole upright (Barto et al., 1983), 20 seeds. |
| **interaction** | 10k episodes, record average regret. |
| **score** | regret normalized [random, optimal] $\rightarrow$ [0,1] |
| **issues** | basic, credit assignment, generalization |

### A.1.5 MOUNTAIN CAR

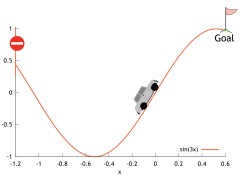

| component | description |
|---|---|
| environments | Agent drives an underpowered car up a hill (Moore, 1990), 20 seeds. |
| interaction | 10k episodes, record average regret. |
| score | regret normalized [random, optimal] $\rightarrow$ [0,1] |
| issues | basic, credit assignment, generalization |

### A.2 STOCHASTICITY

To investigate the robustness of RL agents to noisy rewards, we repeat the experiments from Section A.1 under differing levels of Gaussian noise. This time we allocate the 20 different seeds across 5 levels of Gaussian noise $N(0, \sigma^2)$ for $\sigma = [0.1, 0.3, 1, 3, 10]$ with 4 seeds each.

### A.3 PROBLEM SCALE

To investigate the robustness of RL agents to problem scale, we repeat the experiments from Section A.1 under differing reward scales. This time we allocate the 20 different seeds across 5 levels of reward scaling, where we multiply the observed rewards by $\lambda = [0.01, 0.1, 1, 10, 100]$ with 4 seeds each.

### A.4 EXPLORATION

As an agent interacts with its environment, it observes the outcomes that result from previous states and actions, and learns about the system dynamics. This leads to a fundamental tradeoff: by exploring poorly-understood states and actions the agent can learn to improve future performance, but it may attain better short-run performance by exploiting its existing knowledge. Exploration is the challenge of prioritizing useful information for learning, and the experiments in this section are designed to necessitate efficient exploration for good performance.

### A.4.1 DEEP SEA

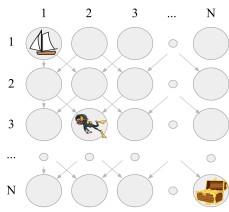

| component | description |
|---|---|
| environments | Deep sea chain environments size N=[5..50]. |
| interaction | 10k episodes, record average regret. |
| score | % of runs with ave regret < 90% random |
| issues | exploration |

### A.4.2 STOCHASTIC DEEP SEA

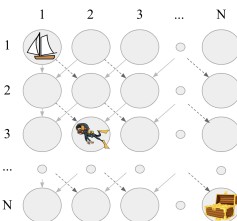

| component | description |
|---|---|
| environments | Deep sea chain environments with stochastic transitions, N(0,1) reward noise, size N=[5..50]. |
| interaction | 10k episodes, record average regret. |
| score | % of runs with ave regret < 90% random |
| issues | exploration, stochasticity |

### A.4.3 Cartpole swingup

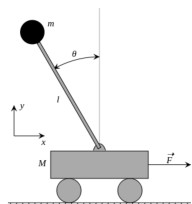

| component | description |
|---|---|
| **environments** | Cartpole 'swing up' problem with sparse reward (Barto et al., 1983), heigh limit x=[0, 0.5, .., 0.95]. |
| **interaction** | 1k episodes, record average regret. |
| **score** | % of runs with average return $> 0$ |
| **issues** | exploration, generalization |

## A.5 Credit assignment

Reinforcement learning extends contextual bandit decision problem to allow long term consequences in decision problems. This means that actions in one timestep can effect dynamics in future timesteps. One of the challenges of this setting is that of *credit assignment*, and the experiments in this section are designed to highlight these issues.

### A.5.1 Umbrella length

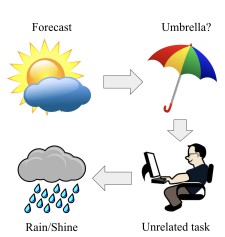

| component | description |
|---|---|
| **environments** | Stylized 'umbrella problem', where only the first decision matters and long chain of confounding variables. Vary length 1..100 logarithmically. |
| **interaction** | 1k episodes, record average regret. |
| **score** | regret normalized [random, optimal] $\rightarrow$ [0,1] |
| **issues** | credit assignment, stochasticity |

### A.5.2 Umbrella features

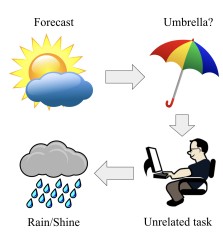

| component | description |
|---|---|
| **environments** | Stylized 'umbrella problem', where only the first decision matters and long chain of confounding variables. Vary features 1..100 logarithmically. |
| **interaction** | 1k episodes, record average regret. |
| **score** | regret normalized [random, optimal] $\rightarrow$ [0,1] |
| **issues** | credit assignment, stochasticity |

### A.5.3 Discounting chain

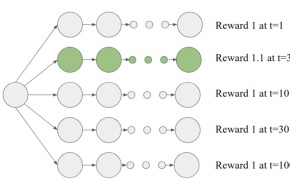

| component | description |
|---|---|
| **environments** | Experiment designed to highlight issues of discounting horizon. |
| **interaction** | 1k episodes, record average regret. |
| **score** | regret normalized [random, optimal] $\rightarrow$ [0,1] |
| **issues** | credit assignment |

## A.6  MEMORY

Memory is the challenge that an agent should be able to curate an effective state representation from a series of observations. In this section we review a series of experiments in which agents with memory can perform much better than those that only have access to the immediate observation.

### A.6.1  MEMORY LENGTH

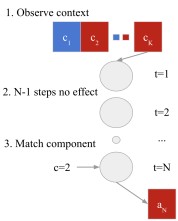

| component | description |
|---|---|
| **environments** | T-maze with a single binary context, grow length 1..100 logarithmically. |
| **interaction** | 1k episodes, record average regret. |
| **score** | regret normalized [random, optimal] $\rightarrow$ [0,1] |
| **issues** | credit assignment |

### A.6.2  MEMORY BITS

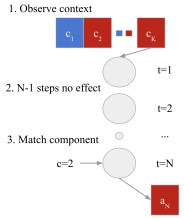

| component | description |
|---|---|
| **environments** | T-maze with length 2, vary number of bits to remember 1..100 logarithmically. |
| **interaction** | 1k episodes, record average regret. |
| **score** | regret normalized [random, optimal] $\rightarrow$ [0,1] |
| **issues** | credit assignment |

## B  BSUITE REPORT AS CONFERENCE APPENDIX

If you run an agent on `bsuite`, and you want to share these results as part of a conference submission, we make it easy to share a single-page 'bsuite report' as part of your appendix. We provide a simple LaTeX file that you can copy/paste into your paper, and is compatible out-the-box with ICLR, ICML and NeurIPS style files. This single page summary displays the summary scores for experiment evaluations for one or more agents, with plots generated automatically from the included ipython notebook. In each report, two sections are left for the authors to fill in: one describing the variants of the agents examined and another to give some brief commentary on the results. We suggest that authors promote more in-depth analysis to their main papers, or simply link to a hosted version of the full `bsuite` analysis online. You can find more details on our automated reports at `github.com/deepmind/bsuite`.

The sections that follow are example `bsuite` reports, that give some example of how these report appendixes might be used. We believe that these simple reports can be a good complement to conference submissions in RL research, that 'sanity check' the elementary properties of algorithmic implementations. An added bonus of `bsuite` is that it is easy to set up a like for like experiment between agents from different 'frameworks' in a way that would be extremely laborious for an individual researcher. If you are writing a conference paper on a new RL algorithm, we believe that it makes sense for you to include a `bsuite` report in the appendix *by default*.

# C  **bsuite** report: benchmarking baseline agents

The *Behaviour Suite for Reinforcement Learning*, or **bsuite** for short, is a collection of carefully-designed experiments that investigate core capabilities of a reinforcement learning (RL) agent. The aim of the **bsuite** project is to collect clear, informative and scalable problems that capture key issues in the design of efficient and general learning algorithms and study agent behaviour through their performance on these shared benchmarks. This report provides a snapshot of agent performance on **bsuite2019**, obtained by running the experiments from **github.com/deepmind/bsuite** (Osband et al., 2019).

## C.1  AGENT DEFINITION

In this experiment all implementations are taken from **bsuite/baselines** with default configurations. We provide a brief summary of the agents run on **bsuite2019**:

- **random**: selects action uniformly at random each timestep.
- **dqn**: Deep Q-networks (Mnih et al., 2015b).
- **boot_dqn**: bootstrapped DQN with prior networks (Osband et al., 2016; 2018).
- **actor_critic_rnn**: an actor critic with recurrent neural network (Mnih et al., 2016).

## C.2  SUMMARY SCORES

Each **bsuite** experiment outputs a summary score in [0,1]. We aggregate these scores by according to key experiment type, according to the standard analysis notebook. A detailed analysis of each of these experiments may be found in a notebook hosted on Colaboratory **bit.ly/bsuite-agents**.

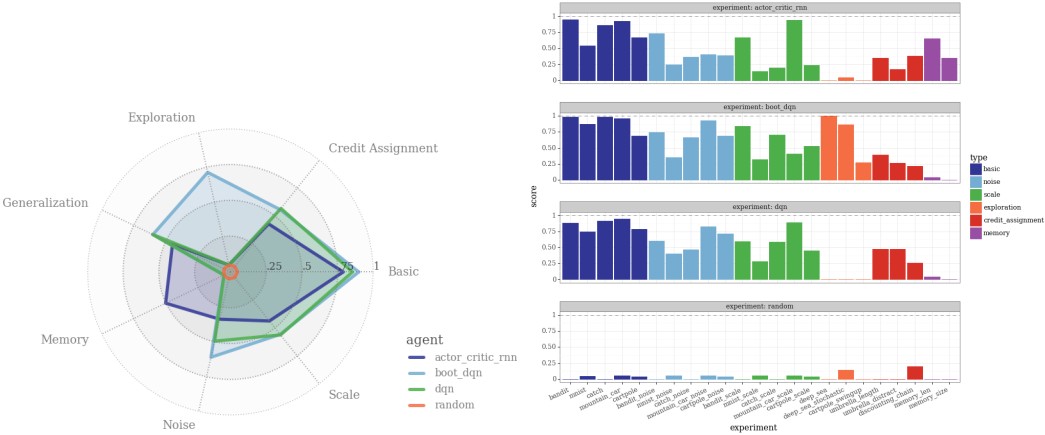

Figure 6: A snapshot of agent behaviour.  Figure 7: Score for each **bsuite** experiment.

## C.3  RESULTS COMMENTARY

- **random** performs uniformly poorly, confirming the scores are working as intended.
- **dqn** performs well on basic tasks, and quite well on credit assignment, generalization, noise and scale. DQN performs extremely poorly across memory and exploration tasks. The feedforward MLP has no mechanism for memory, and $\epsilon$=5%-greedy action selection is inefficient exploration.
- **boot_dqn** is mostly identically to DQN, except for exploration where it greatly outperforms. This result matches our understanding of Bootstrapped DQN as a variant of DQN designed to estimate uncertainty and use this to guide deep exploration.
- **actor_critic_rnn** typically performs worse than either DQN or Bootstrapped DQN on all tasks apart from memory. This agent is the only one able to perform better than random due to its recurrent network architecture.

# D    `bsuite` report: optimization algorithm in DQN

The *Behaviour Suite for Reinforcement Learning*, or `bsuite` for short, is a collection of carefully-designed experiments that investigate core capabilities of a reinforcement learning (RL) agent. The aim of the `bsuite` project is to collect clear, informative and scalable problems that capture key issues in the design of efficient and general learning algorithms and study agent behaviour through their performance on these shared benchmarks. This report provides a snapshot of agent performance on `bsuite2019`, obtained by running the experiments from `github.com/deepmind/bsuite` (Osband et al., 2019).

## D.1    Agent definition

All agents correspond to different instantiations of the DQN agent (Mnih et al., 2015b), as implemented in `bsuite/baselines` but with differnet optimizers from Tensorflow (Abadi et al., 2015). In each case we tune a learning rate to optimize performance on 'basic' tasks from {1e-1, 1e-2, 1e-3}, keeping all other parameters constant at default value.

- **sgd**: vanilla stochastic gradient descent with learning rate 1e-2 (Kiefer & Wolfowitz, 1952).
- **rmsprop**: RMSProp with learning rate 1e-3 (Tieleman & Hinton, 2012).
- **adam**: Adam with learning rate 1e-3 (Kingma & Ba, 2015).

## D.2    Summary scores

Each `bsuite` experiment outputs a summary score in [0,1]. We aggregate these scores by according to key experiment type, according to the standard analysis notebook. A detailed analysis of each of these experiments may be found in a notebook hosted on Colaboratory: `bit.ly/bsuite-optim`.

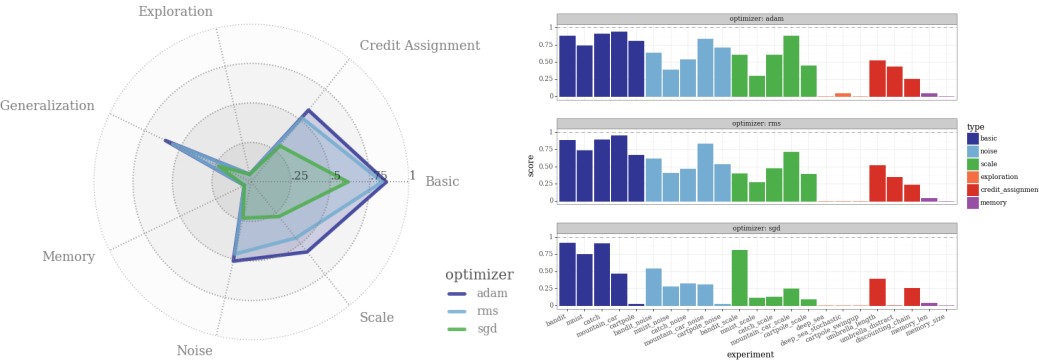

Figure 8: A snapshot of agent behaviour.    Figure 9: Score for each `bsuite` experiment.

## D.3    Results commentary

Both RMSProp and Adam perform better than SGD in every category. In most categories, Adam slightly outperforms RMSprop, although this difference is much more minor. SGD performs particularly badly on environments that require generalization and/or scale. This is not particularly surprising, since we expect the non-adaptive SGD may be more sensitive to learning rate optimization or annealing.

In Figure 11 we can see that the differences are particularly pronounced on the cartpole domains. We hypothesize that this task requires more efficient neural network optimization, and the non-adaptive SGD is prone to numerical issues.

# E    `bsuite` report: ensemble size in Bootstrapped DQN

The *Behaviour Suite for Reinforcement Learning*, or `bsuite` for short, is a collection of carefully-designed experiments that investigate core capabilities of a reinforcement learning (RL) agent. The aim of the `bsuite` project is to collect clear, informative and scalable problems that capture key issues in the design of efficient and general learning algorithms and study agent behaviour through their performance on these shared benchmarks. This report provides a snapshot of agent performance on `bsuite2019`, obtained by running the experiments from `github.com/deepmind/bsuite` (Osband et al., 2019).

## E.1    Agent definition

In this experiment, all agents correspond to different instantiations of a Bootstrapped DQN with prior networks (Osband et al., 2016; 2018). We take the default implementation from `bsuite/baselines`. We investigate the effect of the number of models used in the ensemble, sweeping over {1, 3, 10, 30}.

## E.2    Summary scores

Each `bsuite` experiment outputs a summary score in [0,1]. We aggregate these scores by according to key experiment type, according to the standard analysis notebook. A detailed analysis of each of these experiments may be found in a notebook hosted on Colaboratory: `bit.ly/bsuite-ensemble`.

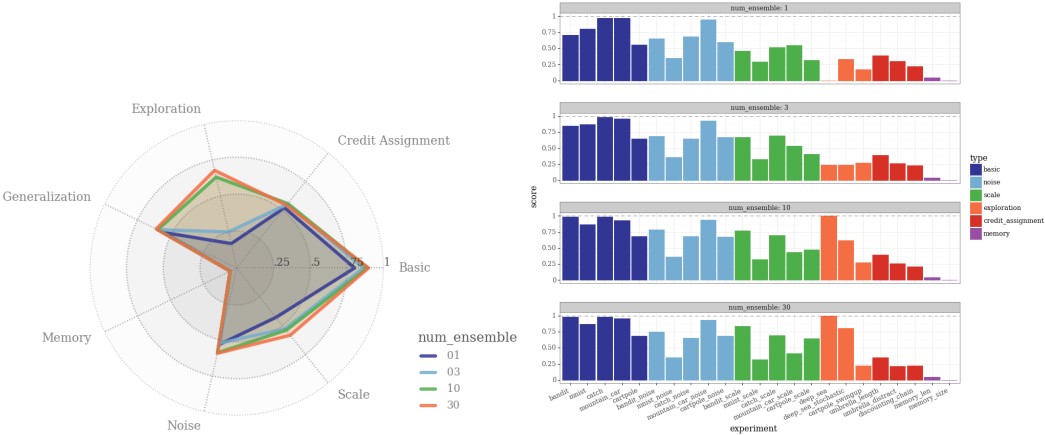

Figure 10: A snapshot of agent behaviour.  Figure 11: Score for each `bsuite` experiment.

## E.3    Results commentary

Generally, increasing the size of the ensemble improves `bsuite` performance across the board. However, we do see signficantly decreasing returns to ensemble size, so that ensemble 30 does not perform much better than size 10. These results are not predicted by the theoretical scaling of proven bounds (Lu & Van Roy, 2017), but are consistent with previous empirical findings (Osband et al., 2017; Russo et al., 2017). The gains are most extreme in the exploration tasks, where ensemble sizes less than 10 are not able to solve large 'deep sea' tasks, but larger ensembles solve them reliably.

Even for large ensemble sizes, our implementation does not completely solve every cartpole swingup instance. Further examination learning curves suggests this may be due to some instability issues, which might be helped by using Double DQN to combat value overestimation (van Hasselt et al., 2016).

