# OpenReview forum: "Behaviour Suite for Reinforcement Learning"
_ICLR.cc/2020/Conference — Accept (Spotlight)_

### Official Review · AnonReviewer3 · 2019-10-21
**Official Blind Review #3**

**Rating:** 6

**Review:**

In this paper, the authors propose a set of benchmarks for evaluating different aspects of reinforcement learning algorithms such as generalisation, exploration, and memory. The aim is to provide a set of simple environments to better understand the RL algorithms and also to provide a set of scores that summarise the performance in each respect. The code of the benchmark is also released.

The paper is well written and clear, and generally can provide a useful contribution. In particular, I like the idea of having a set of benchmarks which can be used for the diagnosis of RL algorithms. Having said this, I have the following concerns which are mostly related to the presentation of the paper. Given clarifications in an author response, I would be willing to increase the score.

- Based on section 1.1 and elsewhere, it seems that the main driver for developing this benchmark has been connecting theory to practical algorithms (which in my opinion is an important step). However, how this can be achieved using the proposed benchmark is not shown in the paper. This can be for example showing how the generalisation score proposed here is linked to theoretical accounts. Or for example in section 2.1, by showing that the memory length 30 for RNN is related to the theoretical expectations. Alternatively, if linking theory and experiments is not the main driver of this work, then it seems a bit unclear what the point of presenting section 1.1 (and other related discussions) is within the context of the paper.

- In terms of novelty, currently the differences between the current work and previous attempts to develop benchmarks is unclear (some examples are mentioned below). In general, a related work section is vital here, but missing in the paper. It should clearly state what the previous attempts in developing benchmarks are, their shortcomings, and how the current work addresses them.

- Some statements in the paper sound more like opinions (which I happen to agree with) rather than something being based on the results of the paper. For example, "We should not turn away from deep RL just because our current theory is not yet developed". It is unclear how this statement is related to the results obtained in this work.

- In section 3, I would like to see some real examples in which bsuite can be used for diagnosis. I find this application of bsuite (diagnosis) very interesting, but as it stands section 3 is more like a tutorial rather than providing a concrete example.

- There are some aspects of RL which are specific to certain classes of RL. For example, in model-based RL, aspects such as the dynamics bottleneck and the planning horizon dilemma have been previously looked at, but are not presented in bsuite. How do the authors envision incorporating such aspects into their framework?

Minor:
- "anything for length ¿ 1" -> replace ¿

- what is the dashed grey line in Fig 4b?

References:
Duan, Yan, et al. "Benchmarking deep reinforcement learning for continuous control." International Conference on Machine Learning. 2016.

Benchmarking Model-Based Reinforcement Learning, Wang et al, 2019.


**Experience Assessment:**

I have read many papers in this area.

**Review Assessment: Checking Correctness Of Derivations And Theory:**

N/A

**Review Assessment: Checking Correctness Of Experiments:**

I assessed the sensibility of the experiments.

**Review Assessment: Thoroughness In Paper Reading:**

N/A

---

> ### Author Response · Authors · 2019-11-07
> **Thank you for your review: we hope our revision will address your concerns**
>
> Thank you very much for your review, we hope our revision will address your concerns.
>
> Q1 - Relating performance to theoretical accounts
>
> We have added a clarification that our RNN agent was implemented with backprop through time of exactly 30 timesteps, so that the sharp performance transition is exactly evidence of this theory <-> practice interplay.
> We had intended to make this clear in the paper, but somehow had forgot to make this explicit.
>
> Similarly, the results of 2.2 are designed to highlight the role of theory-inspired algorithms that outperform the 2^N bound for dithering approaches to exploration.
> We have clarified the meaning of this dashed line at 2^N to try to make this interplay more clear.
>
>
> Q2 - Novelty
>
> Based on your feedback, we have added Section 1.4 on Related Work.
> Here we make a better effort to place bsuite in the context of other benchmarks in RL, and to highlight the specific novelty that we offer.
>
> We believe The Behaviour Suite for Reinforcement Learning offers a complementary approach to existing benchmarks in RL, with several novel components:
>
> - bsuite experiments enforce a specific methodology for agent evaluation beyond just the environment definition.
> This is crucial for scientific comparisons and something that has become a major problem for many benchmark suites (Section 2).
>
> - bsuite aims to isolate core capabilities with targeted `unit tests', rather than integrate general learning ability.
> Other benchmarks evolve by increasing complexity, bsuite aims to remove all confounds from the core agent capabilities of interest (Section 3).
>
> - bsuite experiments are designed with an emphasis on scalability rather than final performance.
> Previous `unit tests' (such as `Taxi' or `RiverSwim') are of fixed size, bsuite experiments are specifically designed to vary the complexity smoothly (Section 2).
>
> - Our open source code has an extraordinary emphasis on the ease of use, and compatibility with RL agents not specifically designed for bsuite
> Evaluating an agent on bsuite is practical even for agents designed for a different benchmark (Section 4).
>
>
> Q3 - Opinionated statements
>
> We agree that, at points, we are making an opinionated case for the value of a certain style of research.
> We hope that we make this clear that it is meant as a *complementary* approach, and that this does not hurt the clarity of our message.
> If there are particular lines you would prefer us to remove or rewrite (potentially that one you highlight) then we will be happy to do this.
>
>
> Q4 - Example analyses
>
> We have included several example bsuite analyses but, in the interests of space, we have relegated these to the Appendices C,D,E.
>
> These include:
>   C - A comparison of DQN, Actor Critic RNN, Bootstrapped DQN and Random agent
>   D - A comparison of "optimizer" algorithm in DQN (SGD, Adam, RmsProp)
>   E - A comparsion of ensemble size in Bootstrapped DQN (size=1, 3, 10, 30)
>
> We hope that these can provide examples of how bsuite can drive interesting research.
> We have edited the section to make these analyses more prominent.
>
>
> Q5 - Other aspects of RL
>
> It is clear that our release of bsuite does not cover all the interesting questions in RL.
> Our aim is to set up a tool that covers *some* interesting diagnostic tools, with the aim of collecting as many as possible of the *best* experiments going forward.
>
> We would love to incorporate excellent experiments that instatiate the dynamics bottleneck, planning horizon and more... we just write an excellent experiment that really measures this well.
> If you, or anyone else, can submit this to the bsuite.committee@gmail.com (or even via github pull) then we would be able to incorporate this very easily.
>
>
> Minor:
> We have taken these into account .
>
>
> Many thanks!

---

> > ### Author Response · Authors · 2019-11-07
> > **Updating review score**
> >
> > (forgot to mention this above)
> >
> > We hope that our revision + response is able to answer your concerns... or if not, please know that we are eager to do this in a secondary revision.
> >
> > If it is enough, then we hope that you will be happy to upgrade your score.
> >
> > Many thanks

---

> > > ### Comment · AnonReviewer3 · 2019-11-11
> > > **Update after author response**
> > >
> > > Thank you for your response. It clarified some of the concerns that I had, so I changed the score. I still think the paper can be improved by including an example of how bsuite can be used for diagnosis purposes and also by revising the opinionated statements which are not directly related to the purpose of the paper (such as the one I mentioned in my earlier comment).

---

> > > > ### Author Response · Authors · 2019-11-13
> > > > **Clarifying requests**
> > > >
> > > > Thank you again for your engagement.
> > > >
> > > > ## Examples of bsuite for diagnosis
> > > >
> > > > My belief is that we have already included three separate examples of how to use bsuite for diagnosis in Appendices C,D,E
> > > >
> > > >   C - A comparison of DQN, Actor Critic RNN, Bootstrapped DQN and Random agent
> > > >   D - A comparison of "optimizer" algorithm in DQN (SGD, Adam, RmsProp)
> > > >   E - A comparsion of ensemble size in Bootstrapped DQN (size=1, 3, 10, 30)
> > > >
> > > > These show how running on bsuite you can get a snapshot of the agent performance across these targeted dimensions.
> > > > Other common examples would include "reimplementing" a baseline agent, and then trying to compare the performance that you would expect to obtain.
> > > >
> > > > Are there other types of examples that you feel we should include?
> > > >
> > > >
> > > > ## Opinionated statements
> > > >
> > > > We believe that part of the value of this paper is in arguing for a principled, focused methodology for research into the core issues in RL research.
> > > > It is clear that this specific sentence was not successful in your review, so we are happy to remove it.
> > > >
> > > > Many thanks

---

### Official Review · AnonReviewer1 · 2019-10-22
**Official Blind Review #1**

**Rating:** 3

**Review:**

Behaviour Suite for Reinforcement Learning

In this paper the authors provide a set of light-weighted but dedicated designed environments, so that researchers can use the environments as a quick indication of the ability of the proposed (or existing) algorithms.
I think the paper is well-written, with the intuition clearly demonstrated.

I tend to vote for rejection though, given that the novelty in the project is relatively limited.
But I believe in general it is a very valuable project that will be beneficial to future research and I would like to recommend for a workshop publication.

Pros:
- The paper is well written, easy to understand.
- Provide an industry level code base that can be used efficiently and easily.
The project will be of great value to the research community in the near future.

Cons:
- The novelty of the project is relatively limited.
The proposed and implemented environments have been studied before.
- No explicit conclusion from the evaluation.


**Experience Assessment:**

I have published in this field for several years.

**Review Assessment: Checking Correctness Of Derivations And Theory:**

I carefully checked the derivations and theory.

**Review Assessment: Checking Correctness Of Experiments:**

I carefully checked the experiments.

**Review Assessment: Thoroughness In Paper Reading:**

I read the paper thoroughly.

---

> ### Author Response · Authors · 2019-11-07
> **Thank you for your review: we hope that our revision will make the value proposition more clear**
>
> Thank you very much for your review!
>
> Based on your feedback we have added Section 1.4, which outlines the relation to prior work more explicitly.
> We also hope that this section makes the *novelty* of our project much more clear.
>
> We believe The Behaviour Suite for Reinforcement Learning offers a complementary approach to existing benchmarks in RL, with several novel components:
>
> - bsuite experiments enforce a specific methodology for agent evaluation beyond just the environment definition.
> This is crucial for scientific comparisons and something that has become a major problem for many benchmark suites (Section 2).
>
> - bsuite aims to isolate core capabilities with targeted `unit tests', rather than integrate general learning ability.
> Other benchmarks evolve by increasing complexity, bsuite aims to remove all confounds from the core agent capabilities of interest (Section 3).
>
> - bsuite experiments are designed with an emphasis on scalability rather than final performance.
> Previous `unit tests' (such as `Taxi' or `RiverSwim') are of fixed size, bsuite experiments are specifically designed to vary the complexity smoothly (Section 2).
>
> - Our open source code has an extraordinary emphasis on the ease of use, and compatibility with RL agents not specifically designed for bsuite
> Evaluating an agent on bsuite is practical even for agents designed for a different benchmark (Section 4).
>
>
> Overall, we are delighted that you agree:
> - The paper is well written, easy to understand.
> - Provide an industry level code base that can be used efficiently and easily.
> - The project will be of great value to the research community in the near future.
>
> We believe that, following the changes we have made to the paper, there might be good reason for you to change your review to an "accept".
>
> Many thanks

---

### Official Review · AnonReviewer2 · 2019-10-23
**Official Blind Review #2**

**Rating:** 8

**Review:**

This paper presents the « Behavior Suite for Reinforcement Learning » (bsuite), which is a set of RL tasks (called « experiments ») meant to evaluate an algorithm’s ability to solve various key challenges in RL. Importantly, these experiments are designed to run fast enough that one can benchmark a new algorithm within a reasonable amount of time (and money). They can thus be seen as a « test suite » for RL, limited to small toy problems but very useful to efficiently debug RL algorithms and get an overview of some of their key properties. The paper describes the motivation behind bsuite, shows detailed results from some classical RL algorithms on a couple of experiments, and gives a high-level overview of how the code is structured.

I really believe such a suite of RL tasks can indeed be extremely useful to RL researchers developing new algorithms, and as a result I would like to encourage this initiative and see it published at ICLR to help it gain additional traction within the RL community.

The paper is easy to read, motivates well the reasons behind bsuite, and shows some convincing examples. However, in my opinion there remain a few important issues with this submission:

1.	There is no « related work » section to position bsuite within the landscape of RL benchmarks (ex: DMLab, ALE / MinAtar, MuJoCo tasks, etc.). I believe it is important to add one.

2.	The current collection of experiments appears to be quite limited. The authors acknowledge the lack of hierarchical RL, but what about other aspects like continuous control, parameterized actions, multi-agent, state representation learning, continual learning, transfer learning, imitation learning / inverse RL, self-play, etc? It is unclear to me whether the goal is to grow bsuite in all these directions (and more) over time, or if there is some kind of « boundary » the authors have in mind regarding the scope of bsuite. Regardless, the fact is that in its current form, bsuite appears to be suited only to a limited subset of current RL research.

3.	I wish an anonymized version of the code had been provided, so that reviewers could test it. In particular I wonder (a) if it is easy to setup and run under Windows, and (b) if it is straighforward to plug a bsuite experiment within an algorithm based on the popular OpenAI gym API (I think the latter is true from what is said at the end of Section 4, but I would have appreciated being able to try it out myself).

Additional minor remarks:
•	I noticed two anoymity-related issues with the provided links: (1) the Google Colab notebook revealed to me the name of its author when clicking the « Open in Playground » link to be able to run it, and (2) the bsuite-tutorial link asks for permission, which might let the authors access reviewer info. I would not hold it against the authors though as I believe these are genuine mistakes and they did their best to preserve anonymity.
•	t > 2 in Section 2.1 should probably be t >= 2
•	In FIg. 2b the label for the y axis seems incorrect since good results are near 0
•	Please explain what is the dashed grey line in Fig. 4b
•	I was unable to understand the last 2 sentences of Section 4
•	Sections C.2, D.2 and E.2 all have the same plots
•	A few typos: incomplete sentence near bottom of p.3 (« the internal workings… »), « These assessment », « expeirments », « recurrant », « length ? 1 », « together with an analysis parses this data », « anonimize », « bsuite environments by implementing », « even if require »

Review update: the authors have addressed my concerns, and I look forward to using bsuite in my research => review score increased to "Accept"

**Experience Assessment:**

I have read many papers in this area.

**Review Assessment: Checking Correctness Of Derivations And Theory:**

N/A

**Review Assessment: Checking Correctness Of Experiments:**

I carefully checked the experiments.

**Review Assessment: Thoroughness In Paper Reading:**

I read the paper thoroughly.

---

> ### Author Response · Authors · 2019-11-07
> **Thank you for your review - we hope to address your main concerns with this revision**
>
> We thank you for your time and comments.
> To address your main concerns:
>
>
> 1 - Related work
>
> This was a clear omission in paper and we have now added a section 1.4 on related work that hopefully clarifies some of these issues.
> If this discussion is still insufficient then we are absolutely happy to improve this further.
>
> One small thing to note is that we want to emphasize these bsuite "experiments" (or tasks) are more than just the RL environment... since they include both the interaction and the analysis.
>
>
> 2 - Limited scope
>
> We hope to grow the bsuite to incorporate as many *excellent* experiments for core RL capabilities as possible.
> However,  we also anticipate that many of the most difficult problems in RL will remain too complex to distill to a simple bsuite example.
>
> Our goal is to collect the best simple, diagnostic tests of Core RL capabilities.
> All of the examples that you list are potential candidates for inclusion *if* we can make an excellent experiment that captures some essence of that problem.
> However, we erred on the side of including *less* where we were unsure of whether we could really make an excellent bsuite experiment.
>
> However, even if the set of bsuite tasks remains relatively simple, it might still play a useful role in the *other* parts of RL research where we don't currently have good experiments.
> A researcher interested in multi-agent RL might still gain some value when running their agent on bsuite... even if it currently does not include a specifically multi-agent experiment.
>
>
> 3 - Anonymized code
>
> We actually submitted an anonymized version of the code together with the initial paper submission. It is linked at the top of this page:
> https://anonymous.4open.science/r/0a9b6721-69c6-42d6-b587-401e0898bfc8/
>
> The confusion may have come from our paper where the link says "github.com/anon/bsuite" but actually clicking the link would also have taken you to that address.
>
> We believe that you will find both the answers (a) and (b) to be positive.
>
>
> Minor remarks:
> - We have taken each of these into account and made appropriate changes to the paper
>
> - Fig 2b grey line represents a baseline 2^N learning time, or a baseline scaling for agents that do not perform deep exploration.
>
> - We rewrote the last sentences of Section 4. This may make more sense when looking at the code, where we have a more explicit example of asking for bsuite environments with custom OBSERVATION_SPEC.
>
>
> Overall we are happy that you agreed with us on the value of our submission.
> We hope that, through our revision, we are able to satisfy your remaining concerns and convert your score to an "accept".
>
> Many thanks

---

> > ### Author Response · Authors · 2019-11-08
> > **Examples of code with OpenAI Gym + Dopamine**
> >
> > To be more specific regarding examples of using OpenAI Gym or Dopamine Frameworks:
> >
> > OpenAI DQN:
> > https://anonymous.4open.science/repository/0a9b6721-69c6-42d6-b587-401e0898bfc8/bsuite/baselines/openai_dqn/run.py
> >
> > OpenAI PPO:
> > https://anonymous.4open.science/repository/0a9b6721-69c6-42d6-b587-401e0898bfc8/bsuite/baselines/openai_ppo/run.py
> >
> > Dopamine DQN:
> > https://anonymous.4open.science/repository/0a9b6721-69c6-42d6-b587-401e0898bfc8/bsuite/baselines/dopamine_dqn/run.py
> >
> >
> > Getting set up on Windows should actually be very simple.
> > We also provide example launch scripts for running on Google Cloud in the README.md section "Running experiments on Google Cloud Platform"
> >
> > The inlcluded scripts provide a step by step way to run any of our baseline agents immediately.
> > This means that it should be possible to prototype an agent in Colab, then run the whole sweep via GCP.
> >
> >
> > Many thanks

---

> > > ### Comment · AnonReviewer2 · 2019-11-12
> > > **Re: Examples of code with OpenAI Gym + Dopamine**
> > >
> > > Thank you!
> > >
> > > Regarding Windows support, the "pip install -e bsuite" worked well, but I ran into issues trying to run the example scripts, between sonnet not being supported under Windows, not having mujoco, or some code not being TF2-compatible... so not as straightforward as one might have hoped.
> > >
> > > That being said, I had a look at the example scripts you mentioned and they are in line with what I was hoping for in terms of quality / simplicity.

---

> > > > ### Author Response · Authors · 2019-11-12
> > > > **Windows support + "baselines" vs bsuite**
> > > >
> > > > We are very glad to hear that the scripts live up to your expectation!
> > > >
> > > > Regarding difficulties running code on Windows... my understanding is that this should not be the case... but obviously offering tech support via anonymous review is a difficult proposal.
> > > >
> > > > I think it's important to separate the core bsuite code, which should work just fine as you say via the pip install, from the "baseline" agents that are examples of specific agent implementations, but not core to anyone else using bsuite with their own working agent.
> > > >
> > > > By default, installing via:
> > > > pip install -e bsuite/
> > > > Will *not* include the baseline dependencies... this is a conscious choice since TF versioning/Sonnet can be difficult to manage.
> > > >
> > > > For example  the agents implemented in Sonnet will not work on windows, but there is not much we can do about that:
> > > > https://github.com/deepmind/sonnet/issues/18
> > > >
> > > > If you have a working implementation of an agent, hopefully the example scripts show that it can be very simple to plug this agent into bsuite.
> > > > If you are on Windows and really want to use our baseline agents, we do provide colabs that can allow you to run this without installing anything on your local machine... so maybe that is an option?
> > > >
> > > >
> > > > Overall, we are really glad that you are engaging with this effort and hope that we can continue to make the user experience even more seemless.
> > > > Hopefully these aspects of the paper will make you keen to recommend our acceptance, either through raising your score, or pushing for acceptance in the post-rebuttal stage.
> > > >
> > > > Many thanks!

---

> > > > > ### Comment · AnonReviewer2 · 2019-11-12
> > > > > **Re: Windows support + "baselines" vs bsuite**
> > > > >
> > > > > Yes indeed this is what I understood (the distinction between bsuite and the baseline agents) -- I just wanted to try and run some quick experiment to see "something", but realized that it was not entirely trivial to run the baseline agents under Windows...
> > > > > Do not worry though, I will not penalize your submission because of it (if I had more time I am sure I could get some of them to run).

---

### Author Response · Authors · 2019-11-13
**Review summary + thanks to the reviewers**

Once again, we would like to thank the reviewers for their efforts and help in improving this paper.

During the review process we have:

1) Added an explicit related work section, which clarifies the position of bsuite with respect to prior work, and the novelty in this project.
2) Clarified links to the opensource code, which reviewers have agreed is generally of high quality.
3) Made the connections between theory and practice more explicit, together with surfacing the "example reports" more clearly in Section 3.

Overall, although there is still one reviewer who tends towards rejection, even that reviewer says:
- The paper is well written, easy to understand.
- Provide an industry level code base that can be used efficiently and easily.
- The project will be of great value to the research community in the near future.

We hope that following our productive discussion with reviewers, together with clarifications on the novelty of this project, that this means the positive aspects of the project shine through enough to recommend acceptance.

Many thanks

---

### Decision · Program_Chairs · 2019-12-19

**Decision:**

Accept (Spotlight)

**Comment:**

This paper proposes a platform for benchmarking and evaluating reinforcement learning algorithms.  While reviewers had some concerns about whether such a tool was necessary given existing tools, reviewers who interacted with the tool found it easy to use and useful. Making such tools is often an engineering task and rarely aligned with typical research value systems, despite potentially acting as a public good. The success or failure of similar tools rely on community acceptance and it is my belief that this tool surpasses the bar to be promoted to the community at a top tier venue.